# Generation and Characterisation of Monoclonal Antibodies against Nairobi Sheep Disease Virus Nucleoprotein

**DOI:** 10.3390/v15091876

**Published:** 2023-09-05

**Authors:** Emmanuel A. Maze, Tiphany Chrun, George Booth, Georgina Limon, Bryan Charleston, Teresa Lambe

**Affiliations:** 1The Pirbright Institute, Ash Road, Pirbright, Woking GU24 0NF, UK; chrun.tiphany@gmail.com (T.C.);; 2Oxford Vaccine Group, Centre for Clinical Vaccinology and Tropical Medicine (CCVTM), Churchill Hospital Old Road, Headington, Oxford OX3 7LE, UK

**Keywords:** Nairobi sheep disease virus, nucleoprotein, mouse monoclonal antibody, *Orthonairovirus*

## Abstract

Nairobi sheep disease (NSD), caused by the viral agent NSD virus (NSDV), is a haemorrhagic fever disease affecting and inducing high mortality in sheep and goat populations. NSDV belongs to the genus *Orthonairovirus* of the *Nairoviridae* family from the order *Bunyavirales*. Other viruses circulating in livestock such as Crimean–Congo haemorrhagic fever virus (CCHFV) and Dugbe virus (DUGV) are members of the same genus and are reported to share antigenic features. There are very few available materials to study NSDV infection both in vitro and in vivo. In the present work, we characterised two monoclonal antibodies generated in mice that recognise NSDV specifically but not CCHFV or DUGV, along with a potential use to define virus-infected cells, using flow cytometry. We believe this tool can be useful for research, but also NSDV diagnostics, especially through immunological staining.

## 1. Introduction

Nairobi sheep disease (NSD) is a viral haemorrhagic fever disease affecting and inducing high mortality in sheep and goat populations [1]. First reported in 1917 from an outbreak in Kenya [2], subsequent outbreaks, including one in Uganda in 1958 [3], have occurred when naïve animals lacking a pre-existing immunity have been moved from areas free from disease occurrence to regions where the disease is endemic. Once infected, a large number of animals are often lost, reaching in some instances ~90% mortality [1]. The aetiological agent, the NSD virus (NSDV), is believed to circulate in East Africa (Kenya, Somalia, Uganda, and Tanzania) and Southern Asia (India, Sri Lanka, and China) [4]. 

In terms of its phylogeny, NSDV is classified within the *Orthonairovirus* genus of the *Nairoviridae* family of the order *Bunyavirales*. The virus is composed of three negative-sense RNA segments: small (S), encoding for the viral nucleoprotein (NP); medium (M), encoding for the viral glycoproteins; and long (L), encoding for the viral polymerase [5]. Importantly, NSDV belongs to the same taxon as Crimean–Congo haemorrhagic fever virus (CCHFV), a zoonotic disease that is a threat to public health due to its ability to cause haemorrhagic fever in humans. CCHFV has a wide reported distribution, including countries in Africa, the Middle East, Southern Asia, and South-Eastern Europe [6]. As a result, CCHFV distribution overlaps with NSDV.

In comparison with CCHFV, there are very few materials to study NSDV infection both in vitro and in vivo. In particular, a tool to monitor NSDV infection through staining of the virus NP, to decipher the viral tropism, would be useful [7]. Additionally, in enzootic areas, a priority lies in distinguishing NSDV from the other orthonairoviruses that co-circulate in such regions. Such a task is complicated due to animals showing non-specific post-mortem clinical signs following an infection. According to the World Organisation of Animal Health (WOAH), the identification of the agent involved requires virus isolation in tissue culture (using a baby hamster kidney BHK21 cell line) followed by antigenic tests to confirm the presence of NSDV, with an immunofluorescence assay using a specific anti-NSDV antiserum being suitable for this purpose [8]. However, a cross-reaction may occur with other members of the genus *Orthonairovirus*, especially Dugbe virus (DUGV) [9,10]. Other methods such as RT-PCR are more specific in distinguishing viruses, although in some instances, cross-amplification with NSDV-specific primers can occur and has led to the discovery of Kupe virus (KUPV), a DUGV-related virus [11].

In the present work, based on the alignment of members of the *Nairoviridae* family, we generate and characterise two monoclonal antibodies against a NSDV NP-derived peptide that displays a low sequence homology to other orthonairoviruses. We go on to demonstrate their ability to recognise NSDV NP without interacting with DUGV and CCHFV NPs. We believe our work provides a source for an anti-NSDV NP monoclonal antibody useful for research on NSDV and diagnostic analysis.

## 2. Materials and Methods

### 2.1. Nairovirus Alignment

An alignment of nairovirus NPs was produced at the amino acid level using the ClustalW algorithm in the software MEGA 7.0. A few members of each *Orthonairovirus* species were aligned together, with the list of sequences used provided in Appendix A. Searching through the NP sequences of aligned viruses, regions of relatively high amino acid divergence (many mismatches in comparison to NSDV strains) were identified, with two candidates selected for mice immunisation. 

### 2.2. Production of Mouse Monoclonal Antibodies

Peptide candidates were submitted to Bioserv UK, which performed monoclonal antibody production in mice. Ethical approval was obtained, with animal handling taken care of by the company, in accordance with United Kingdom regulations. A total of four mice were used (M1-4). Synthetic peptides from protein sequences exclusive to NSDV among orthonairoviruses, linked to carrier proteins such as Keyhole limpet hemocyanin (KLH), were used to hyperimmunise mice subcutaneously in the presence of a suitable adjuvant. Antisera were screened initially against each individual peptide. After termination, splenocytes from two out of four mice (M1 and M2) were isolated and fused with an immortalised cell line to create hybridomas. Such hybridomas were further cloned (limit dilution) and grown to a cell number sufficient to produce antibodies for the screening of the anti-NSDV NP hybridomas. Some of the best candidates, displaying a good response to NSDV peptides, were tested at The Pirbright Institute against NSDV-infected BHK21 cells to confirm reactivity to the antigen of interest. Two NSDV-reactive hybridomas were cultured and antibody scale-up production was performed, producing stock concentrations of 1 mg/mL for each monoclonal antibody.

### 2.3. Cell Lines

BHK21 cells were cultured in complete GMEM made of Glasgow minimum essential medium (GMEM) supplemented with foetal bovine serum (FBS; 10%), penicillin/streptomycin (100U; Cat # P4333, Sigma-Aldrich, Burlington, MA, USA), L-glutamine (2 mM; Cat # 59202C, Sigma-Aldrich), and tryptose phosphate broth (5%; Cat # T8159, Sigma-Aldrich) at 37 °C and 5% CO_2_.

### 2.4. NSDV Isolate and Viral Preparation

The NSDV isolate ND66 PC9 was obtained initially from World Reference [12]. The virus was grown on BHK21 cells for three days before harvesting. Briefly, BHK21 were cultured in 175 cm^2^ flasks until they reached 90–100% confluence. Cells were washed once in PBS, and 15 mL of virus inoculum (1:1000), prepared in FBS- and antibiotic-free GMEM, was applied onto the cells for an hour. The inoculum was then removed, and cells were cultured in 45 mL of complete GMEM containing 2% FBS. Cells were kept for 3 days until cytopathic effects (CPE) appeared. The medium containing virus particles was centrifuged at 3500× *g* for 10 min to remove cellular debris. The supernatant was aliquoted and stored at −80 °C until use. Up to three vials were used to determine the average viral titre in distinct experiments, as described below. All experiments involving the handling of NSDV were performed within the Pirbright Institute’s Specified Animal Pathogens Order (SAPO) 4 containment laboratories.

### 2.5. Viral Titration in Supernatant

The amount of virus released in culture medium was measured as tissue culture infectious dose 50 (TCID_50_). Briefly, viral supernatant was 10-fold serial diluted from 1 to 1:100,000 in 50 μL of serum-free GMEM in a 96-well flat-bottom plate. Six replicates were used per dilution. An amount of 100 μL of complete GMEM containing 5 × 10^3^ BHK21 cells was added per well. Cells were cultured for up to 4 days, monitoring the appearance of CPE. The last dilution at which CPE appeared in all wells and the number of wells displaying CPE in further dilutions were recorded, then the viral titre in TCID_50_/mL was obtained using the Spearman–Karber formula.

### 2.6. Immunofluorescent Assay (IFA)

BHK21 cells were plated at 3 × 10^5^ cells/100 μL/well in a 96-well flat-bottom plate, in the presence of either 50 μL of NSDV at the equivalent of 10^5^ TCID_50_/well or 50 μL of GMEM (uninfected control). The cells were cultured overnight at 37 °C and 5% CO_2_. Following incubation with the virus, the culture medium was discarded, and cells were fixed in the plate with 100 μL/well of cold methanol for 10 min. Cells were then washed three times with PBS before 100 μL/well of 2% BSA in PBS was added to the cells and incubated for an hour (blocking step). After blocking, 50 μL/well of supernatants from hybridoma cultures was added and left for an hour. The cells were washed three times with PBS; then, a secondary goat anti-mouse Alexa Fluor 488-conjugated antibody (Cat # A1101; Thermofisher scientific, Waltham, MA, USA) was used at 1:1000 for an hour. Finally, cells were washed, and staining was visualised using the IncuCyte^®^ S3 Live-Cell Analysis System (Sartorius, Göttingen, Germany). 

### 2.7. Nairobi Sheep Disease Virus Staining via Flow Cytometry

BHK21 cells were plated at 10^6^ cells/well on a 6-well plate. Cells were infected by adding 2 mL of virus per well with a 5-fold decreasing virus titre per well—virus diluted in GMEM at 1:10 to 1:6250—and one well was kept uninfected (GMEM only). The cells were incubated overnight at 37 °C and 5% CO_2_, washed once with PBS, trypsinised, and harvested. Cells were fixed in 100 μL PFA 2% in PBS followed by permeabilisation in Triton X100 (0.1% in PBS), fixed and permeabilised using cytofix/cytoperm reagents (Cat # 555028, BD) according to the manufacturer’s instruction, or fixed and permeabilised in 100 μL cold methanol only, for 10 min. Cells were washed twice in PBS; then, a blocking step in 200 μL of 2% BSA in PBS was performed before an incubation with 50 μL solutions containing company-purified monoclonal antibodies. Cells were washed further in PBS before adding a secondary anti-mouse IgG FITC-conjugated antibody at 1:100. Finally, the cells were washed two times in PBS before being analysed in an LSR Fortessa II flow cytometer. To confirm the antibody isotype, secondary anti-mouse IgG1, IgG2a, or IgG2b were used.

### 2.8. Enzyme-Linked ImmunoSorbent Assay (ELISA)

Recombinant NSDV NP (Accession number: YP_009361831.1) was produced and provided by Reading University; DUGV NP (Accession number: Q8V336; Cat # CSB-EP851800DCAH-CSB) was from Stratech; and CCHFV NP (Accession number: NP_950237; Cat# REC31639) was from the Native Antigen Company, Oxford. Monoclonal antibodies were tested for their ability to bind NSDV, DUGV, and CCHFV NPs using ELISA. Briefly, wells of a Maxisorp plate (Cat # 44-2404-21, Thermo Fisher Scientific) were coated overnight with ~50 ng of protein per well of either NSDV NP, DUGV NP, or CCHFV NP. The plate was washed four times in PBS containing 0.05% Tween20 (PBST), blocked with PBST containing 4% skimmed milk (PBST-4% milk), and incubated for 45 min with monoclonal antibody diluted in PBST-4% milk. The plate was washed four times in PBST, and a secondary goat anti-mouse IgG antibody conjugated to HRP (Cat # A16072; Thermo Fisher Scientific) diluted 1:5000 in PBST-4% milk was added for 45 min. The plate was finally washed four times in PBST, before being developed using TMB (Cat # TMBW-1000-01; Cambridge bioscience, Cambridge, UK), and stopped using Stop solution (Cat # NSTP-1000-01; Cambridge bioscience).

### 2.9. Western Blot

Proteins were separated on a 12% polyacrylamide gel (Cat # 4561046; Biorad, Hercules, CA, USA) and then transferred onto a PVDF membrane (Cat # 1620177; Biorad). Membranes were blocked in PBS containing 10% skimmed milk (PBS-10% milk), followed by an incubation overnight at 4 °C with PBS-10% milk containing primary antibody at 1:1000. Membranes were washed and incubated at room temperature, with 1:5000 secondary anti-mouse HRP-conjugated antibody in PBS-10% milk for at least 1 h. Chemiluminescence was used to reveal blots.

## 3. Results

### 3.1. Alignment of Orthonairoviruses 

There are a limited number of sequences per species of orthonairoviruses in public depositories. The amino acid sequences of a few strains per species were directly collected or translated from a nucleotide sequence (Sapphire II virus, KU343165.1) obtained from GenBank (Appendix A). The rationale was to find an amino acid sequence exclusive to NSDV among members of the same genus. Forty-three sequences from 12 species of orthonairoviruses were aligned, including *Nairobi sheep disease, Dugbe, Hazara, Crimean–Congo haemorrhagic fever, Thiafora, Sakhalin, Burana, Qalyub, Ketarah, Kasokero, Dera Ghazi Khan, and Hughes nairoviruses*. We focused on regions of low sequence homology, i.e., regions of particular interest presenting a lot of mismatches were retained (Figure 1). Overall, two peptide candidates were retained with peptide #1 SGYEVSMRLVSSES (SG…SES) displaying 93–100% identity among NSDV strains and <37% identity to all other *Orthonairovirus* species and peptide #2 PKVAEDLKESLKSLVAWINAH (PR…NAH) displaying 86–100% identity among NSDV strains and <34% identity to all other *Orthonairovirus* species. The closest relative for both peptides was Kupe virus (36% and 33% identity to NSDV for peptide #1 and #2, respectively), a strain of the species *Dugbe nairovirus*. Percentages of identity to each species are presented in Appendix A.

### 3.2. Identification of Positive Hybridoma Clones 

Peptide candidates were submitted for synthesis and mouse immunisation to Bioserv UK. After immunisations, all mice mounted an antibody response to the peptides (data not shown). Two mice (M1 and M2) were sacrificed, hybridomas were generated and sub-cloned in vitro, and their culture supernatants were assessed for the presence of anti-NSDV peptide antibodies. The most promising hybridomas displayed OD values in ELISA > 1 and were further screened for their ability to recognise native NP in NSDV-infected BHK21 cells in vitro using an immunofluorescent assay (IFA, Appendix A). Seven hybridomas displayed a clear positive response to NSDV-infected cells (AC6, AE2, AF5, and GG12 from mouse M1; EG6, FB10, and FH3 from mouse M2). However, only three hybridoma clones, namely GG12, AF5, and AE2, produced a positive staining of NSDV-infected cells (>45%) while having a negative staining of uninfected cells (<0.1%). M1-derived AC6 and M2-derived EG6, FB10, and FH3 reliably stained 32%, 10%, 11%, and 2% of uninfected control cells, respectively. GG12 and AF5 clones were chosen for scale-up production.

### 3.3. Antibody Clones GG12 and AF5 Are Specific to NSDV But Not CCHFV 

We sought to test whether the monoclonal antibody clones GG12 and AF5 that were generated were specific to NSDV and not to other members of the same genus. To do so, an ELISA was initially performed, using recombinant NPs from NSDV, DUGV, and CCHFV as coating antigens, respectively. An absorbance signal (blanked O.D. > 1) was observed for NSDV NP, but not for either DUGV or CCHFV NPs (Figure 2). Secondly, a Western blot was performed including recombinant NPs from NSDV, DUGV, and CCHFV. As shown in Figure 2, the GG12 monoclonal antibody was able to react with NSDV NP, but not DUGV and CCHFV NPs. Unfortunately, due to technical issues; we have not performed a Western blot analysis for AF5. Nonetheless, these observations suggest that we were successful in producing monoclonal antibodies that do not bind the NP antigen from other genus members such as DUGV and CCHFV and, hence, may be exclusive to NSDV.

### 3.4. Staining of Infectious Cells through Flow Cytometry 

We further assessed whether the monoclonal antibodies could be used in flow cytometry-related applications to estimate the amount of cells that can be infected in vitro. For this purpose, BHK21 cells were infected with NSDV at varying virus dilutions for more than 18 h (overnight), before being fixed and permeabilised following PFA-Triton X100, cytofix/cytoperm, or methanol treatments (see methods). The cells were stained with GG12 or AF5 antibodies, followed by a secondary anti-mouse IgG Alexa Fluor 488-conjugated antibody. At the highest virus concentration, the PFA-Triton X100 and cytofix/cytoperm staining approach yielded an overall 12–15% and 0–2%, respectively (Figure 3a). Conversely, cells fixed and permeabilised using methanol treatment displayed a high infectious ratio, reaching 80–89% at the highest virus concentration (Figure 3b). This suggests that methanol preparation is more adequate at revealing NP peptide epitope bound by GG12 and AF5 monoclonal antibodies. Finally, both antibody clones were found to belong to the IgG1 subclass through flow cytometry, using secondary anti-mouse IgG1, IgG2a, or IgG2b.

## 4. Discussion

The generated anti-NSDV monoclonal antibodies GG12 and AF5 were shown to work well in the IFA test (see Section 3.2) and, as a result, are suitable for assessing the presence of NSDV in samples upon virus isolation in BHK21, in accordance with WOAH guidelines [8]. However, in order to validate the true potential of these monoclonal antibodies, their reactivity must be assessed against tissue samples derived from NSDV-infected sheep to account for any background. 

Furthermore, since NSDV-derived peptides could be used as bait in an ELISA that reacted well with our antibodies directed against NSDV NP, it would be of great interest to assess the reactivity of sera derived from established NSDV-infected sheep that have undergone seroconversion [13]. Such peptides may offer potential support materials for the development of ELISA-based diagnostic tests that are specific to NSDV among other orthonairoviruses. 

Another challenge relies on assessing the presence of co-infecting agents in samples from exposed animals, especially in regions where more than one orthonairovirus circulates [14]. Upon viral isolation, GG12 or AF5 monoclonal antibody used alongside another virus-specific antibody could offer a solution in these situations, provided that samples are collected in the timeframe when both viral agents are replicating within the exposed individuals. Finally, in the case of modelling the co-infection in vitro to assess the effect of such an event on all orthonairoviruses involved, our designated antibodies could allow for precise monitoring of the evolution of NSDV replication among others.

## 5. Conclusions

We successfully generated monoclonal antibodies GG12 and AF5, which are able to detect NSDV-infected cells and to bind NSDV NP, with limited cross-reactivity that we could not detect across other orthonairoviruses. We believe these new tools will facilitate research on the neglected Nairobi sheep disease.

## Figures and Tables

**Figure 1 viruses-15-01876-f001:**
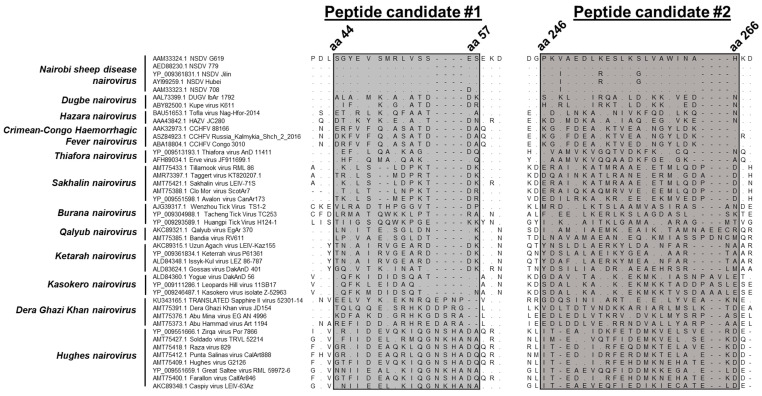
Alignment of peptide candidates across orthonairoviruses. The alignments of peptide #1 (SG…SES, on the left) and #2 (PR…NAH, on the right) of 43 viral sequences from 12 *Orthonairovirus* species. A dot (.) represents a matching amino acid; a dash (-) is for a gap; and another letter different from the one on the first lane is for a mismatch, specifying which amino acid is present instead. Amino acid positions are provided for NSDV. Percentages of mismatches were estimated after removal of gaps from the alignment (Appendix A).

**Figure 2 viruses-15-01876-f002:**
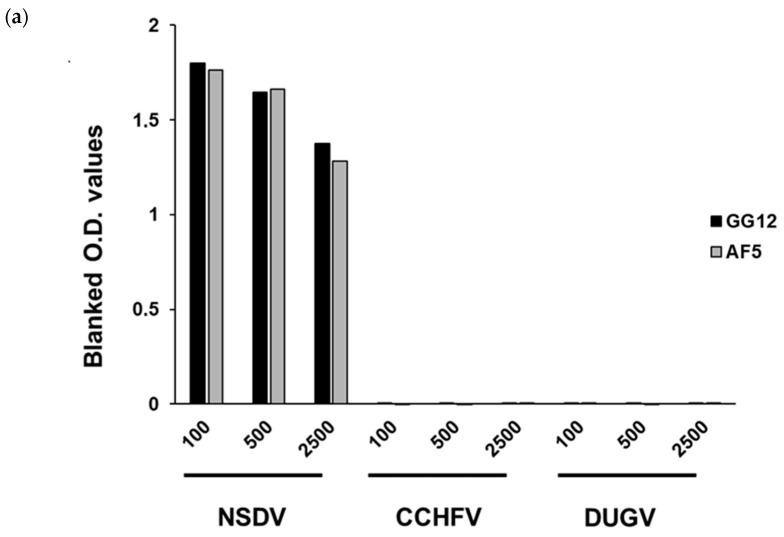
Monoclonal antibodies generated react to NSDV but neither CCHFV nor DUGV NP. GG12 and AF5 were assessed for their ability to bind to NSDV NP but not CCHFV and DUGV NP. (**a**) ELISA performed using either GG12 (black) or AF5 (grey) at 1:100–1:2500 dilution (1 mg/mL stock concentration) on plates coated with 0.50 ng/well of NSDV, CCHFV or DUGV NP. (**b**) Western blot using GG12 at 1:2000 dilution on membranes containing NSDV (first lane), DUGV (middle lane), and CCHFV (right lane). Ponceau staining was performed to confirm successful transfer of all NPs onto the membrane.

**Figure 3 viruses-15-01876-f003:**
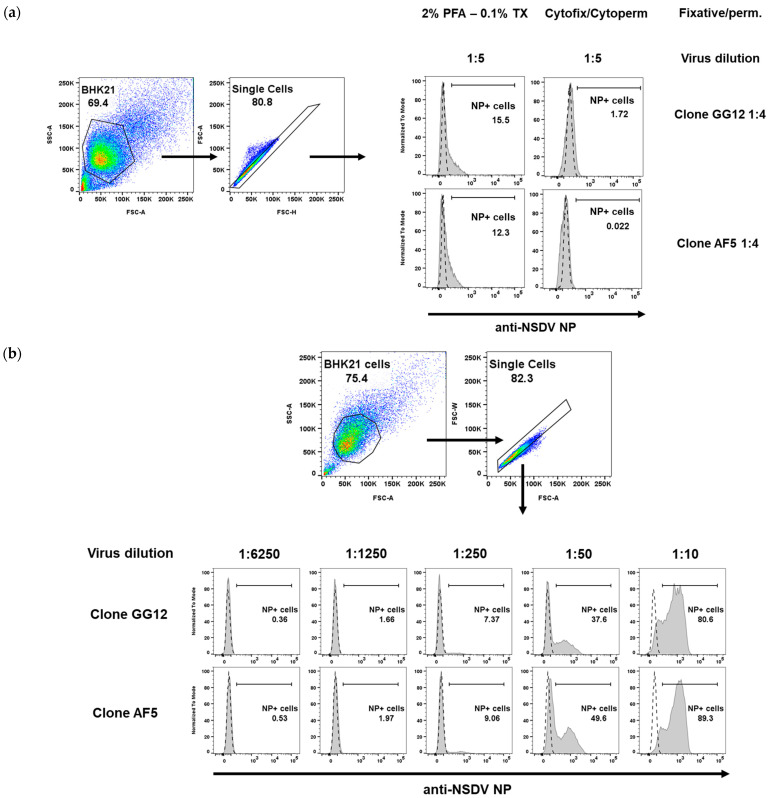
Flow cytometry analysis of NSDV-infected BHK cells. (**a**) PFA/Triton X100 (PFA—0.1% TX) and Cytofix/Cytoperm treatments result in poor staining for both monoclonal antibodies used at a high concentration (1:4) and at a high concentration of virus (dilution of 1:5). (**b**) Methanol fixation-permeabilisation allows for staining of NSDV-infected cells in a dose-dependent manner. Cells were infected with varying dilutions of virus (1:10 to 1:6250) and both antibodies were used at 1:50. For (**a**,**b**), the gating strategy is provided; dashed line in histograms represent NP staining of uninfected BHK cells while grey curve represents NP staining for BHK-infected cells.

## Data Availability

The data supporting the main findings of this study are included in both the main manuscript and the Appendix A.

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
