# Peer review of "Generation and Characterisation of Monoclonal Antibodies against Nairobi Sheep Disease Virus Nucleoprotein"

_viruses, 2023, doi:10.3390/v15091876_

Round 1

Reviewer 1 Report

The article submitted for my evaluation aims to produce immunological reagents (here, monoclonal antibodies) against the Nairobi sheep disease virus (NSDV) present in small ruminants on the African continent. The result presented, ie the selection of two monoclonal antibodies directed against the NSDV nucleoprotein, meets the objective set.

The article is clear and well written; I underlined the average originality of the publication, which in no way diminishes the interest of this important work to fight against this animal disease often neglected despite the impact on livestock.

I would like to add two remarks:

1- it is mentioned "a suitable adjuvant" for the immunization of mice intended to produce monoclonal antibodies; it would be more informative to directly specify the adjuvant used

2- the NSDV being exotic and potentially dangerous for British sheep farming, it would be good to specify in the Materials and Methods that the manipulations of the virus were carried out in a protected level 3 laboratory (BLS3)

Finally, as the authors rightly mentioned in the discussion, the relevance of these two monoclonal antibodies must be tested under field conditions, i.e. in sheep naturally infected with NSDV.

Author Response

The authors would like to thank the reviewer for the reviewing and comments. Regarding the few remarks, please see our reply/amendments below:

1- it is mentioned "a suitable adjuvant" for the immunization of mice intended to produce monoclonal antibodies; it would be more informative to directly specify the adjuvant used

This is a good point. Unfortunately, the private company did not disclose the adjuvant used in their procedure and so we could not add in the information.

2- the NSDV being exotic and potentially dangerous for British sheep farming, it would be good to specify in the Materials and Methods that the manipulations of the virus were carried out in a protected level 3 laboratory (BLS3)

Thanks for the notice. A statement has been included in lines 103-105, specifying the work has been carried under Specified Animal Pathogens Order containment laboratories in place at the Pirbright Institute.

Reviewer 2 Report

Comments on “Generation and characterization of monoclonal antibodies against Nairobi sheep disease virus nucleoprotein” by Maze et al.

Nairobi sheep disease is a serious agricultural scourge caused by NSDV which is transmitted by ticks in East Africa and South Asia.  Because sheep and goat mortality after infection ranges up to 90%, the disease has devastating economic consequences.  Despite its severity, there are few molecular experimental or diagnostic tools available to diagnose and track NSDV infection.  A further complication is that similarity of gross pathology caused by NSDV is similar to that of other overlapping Nairoviruses, making accurate diagnosis of NSDV infection difficult.

This manuscript describes the development and initial testing of the first monoclonal antibodies directed at a relatively unique region of the NSDV nucleoprotein.  The histochemistry, ELISA, flow cytometry, and western analyses show robust reactivity of the new antibodies to NSDV NP but not to the nucleoproteins of other related viruses.

The manuscript is well written and binding and specificity of the antibodies toward NSDV NP are painstakingly documented.  Although the manuscript is very narrow and technical in scope, it represents an important advance necessary to better study NSDV from an epidemiological and molecular virological standpoint.

The only point that seems valuable to further substantiate the utility of these new monoclonal antibodies would be to demonstrate their properties on actual infected sheep samples in addition to purified NP and/or infected cell lines, as the authors note.

Author Response

The authors are thankful for the review and comments.